# Adversarial training with cycle consistency for unsupervised super-resolution in endomicroscopy

**Daniele Ravì**[1], **Agnieszka Barbara Szczotka**[1], **Dzhoshkun Ismail Shakir**[1],
**Stephen P Pereira**[2], **Tom Vercauteren**[3]
[1] Wellcome / EPSRC Centre for Interventional and
Surgical Sciences, University College London
[2]UCL Institute for Liver and Digestive Health
[3]School of Biomedical Engineering
Imaging Sciences, King's College London
d.ravi@ucl.ac.uk

## Abstract

In recent years, endomicroscopy imaging has become increasingly used for diagnostic purposes. It can provide intraoperative aids for real-time tissue characterization and can help to perform visual investigations aimed to discover epithelial cancers. However, accurate diagnosis and correct treatments are partially hampered by the low numbers of informative pixels generated by these devices. In the last decades, progress has been made to improve the hardware acquisition and the related image reconstruction in this domain. Nonetheless, due to the imaging environment, and the associated physical constraints, images with the desired resolution are still difficult to produce. Post-processing techniques, such as Super Resolution (SR), are an alternative solution to increase the quality of these images. SR techniques are often supervised, requiring aligned pairs of low-resolution (LR) and high-resolution (HR) patches to train a model. However, in some domains, the lack of HR images hinders the generation of these pairs and makes supervised training unsuitable. For this reason, we propose an unsupervised SR framework based on an adversarial deep neural network with a physically-inspired cycle consistency, designed to impose some acquisition properties on the super-resolved images. Our framework can exploit HR images, regardless of the domain where they are coming from, to transfer the super-resolution to the initial LR images. This property can be particularly useful in all situations where pairs of LR/HR are not available during the training. Our quantitative analysis, validated using a database of 238 endomicroscopy video sequences, shows the ability of the pipeline to produce convincing super-resolved images. A Mean Opinion Score (MOS) study also confirms this quantitative image quality assessment.

## 1 Introduction

According to a recent report by the World Health Organization, cancer is the second leading cause of death after cardiovascular disease and was responsible for 8.8 million deaths in 2015. Early detection, such as the ability to detect precancerous lesions, plays an important role in reducing cancer incidence and related mortality [15]. Optical endomicroscopy, based for example on confocal microscopy, optical coherence tomography or spectroscopy, has the ability to perform optical biopsies and identify early pathology in tissues or organs including the colon, esophagus, pancreas, bladder, liver and cervix [11]. Although, in the last years, progress has been made to build reliable optical endomicroscopy devices [10], the need to operate at micron scale through the use of endoscopes, fibre bundles, laparoscopes, and needles, limits the final resolution of the images. Further hardware

1st Conference on Medical Imaging with Deep Learning (MIDL 2018), Amsterdam, The Netherlands.

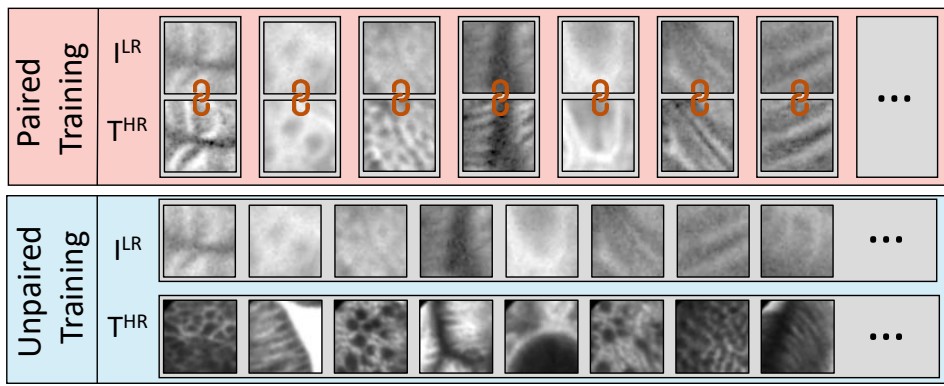

Figure 1: Example of aligned-paired and unpaired patches used for training super-resolution neural networks.

improvements are very difficult to achieve and one possibility to improve the image quality is to post-process them using SR techniques.

Recent methods for SR are based on training models that learn how to improve image resolution by exploiting a database of aligned pairs of LR and HR images. Nonetheless, due to the lack of HR endomicroscopy images, these pairs are not always available in this domain. An option is to generate these pairs synthetically, but this is only feasible when the acquisition process is well defined. In most of the cases, the acquisition process is unknown or hard to define and supervised methods may not be applicable.

For this reason, we designed a deep learning architecture trained in an unsupervised manner where the aforementioned one-to-one alignment between LR and HR is not required anymore. We formalize our framework so that LR images from an initial input domain $I^{LR}$ can be transformed into images of any target domain $T^{HR}$. The target domain can be the same or different from the initial one. An example of the difference between initial and target domain with paired and unpaired patches is shown in Fig. 1.

To allow the use of unpaired patches in the training procedure, we make use of an adversarial network, a class of artificial intelligence algorithms which train two separate models that challenge each other in a zero-sum game. The first model is a $SR$ network that learns how to improve the resolution of the images, and the second is a discriminative network $DS$ that, looking at the target domain, tries to distinguish images generated by the $SR$ network from the real $T^{HR}$ images. The aim of $SR$ is to understand how to fool the $DS$ network and this leads to a generation of better super-resolved images.

Adversarial training can learn how to produce outputs with the same distribution as the target domain. However, the target domain distribution can be matched by simply mapping the input images to any random permutation of images in the target domain. Therefore, without specific constraints, the network could produce arbitrary $T^{HR}$ images with no direct relationship with the input image. In other words, an adversarial loss, alone, cannot guarantee that the learned function maps an input to a desired corresponding super-resolved image. Following the idea proposed by [19], we add in the adversarial training a further component that imposes a consistency between the HR images and the initial LR images. This consistency is obtained by constraining the super-resolved $T^{H\tilde{R}}$ image to have similar physical acquisition properties to the initial $I^{LR}$ image.

To the best of our knowledge, this paper is the first to propose an adversarial network that takes advantage of the knowledge of the physical acquisition process to impose a cycle consistency and perform unsupervised SR of medical images. In our experiment, we show that the proposed framework does not require paired aligned patches for the training. This is an important property in all the domains where HR images are not available. The rest of the paper is organized as follows: Section 2 presents the state-of-the-art for unsupervised SR methods. Section 3 presents the proposed training methodology based on an adversarial training with cycle consistency. Section 4 presents the

results obtained using a quantitative image quality assessment and a Mean Opinion Score (MOS) study and section 5 summarizes the contribution of this research.

## 2 Related work

With the recent outbreak of deep learning, example-based super-resolution (EBSR) has led to a dramatic leap in SR performance [13]. These approaches are mainly based on a supervised training procedure where a database of aligned pairs of LR and HR images is required to create the model. Being supervised, these SR methods are restricted to specific training data, where the LR images are usually predetermined from their HR counterparts. However, in many contexts, such as in endomicroscopy, HR images are not available due to physical constraints and therefore these paired aligned images cannot be generated. A first attempt to train an EBSR network for endomicroscopy, was proposed by [12] where a video-registration technique is used to estimate the HR images from a sequence of LR images. A pipeline for generation of synthetic data is finally presented to produce the desired aligned pairs. Although models trained with generated synthetic data can obtain convincing SR images, the domain gap between reconsecrated HR images and original pCLE images raises questions about their reliability for clinical use. For this reason, we believe that unsupervised super-resolution techniques would be more suitable in these cases. For example, [3] presented an unsupervised method for image SR. The approach uses a Variational Bayesian (VB) algorithm that combines a Bayesian technique with a Markovian model. The main issue with this approach is the difficulty to hand-craft a good perceptual loss function and the final images tend to be blurred. Rather than designing a suitable similarity loss function, [5] proposed a general framework called Generative Adversarial Network (GAN) where the perceptual loss function is trained directly using a discriminative network. This allows the method to automatically verify if a generated sample is similar to the real one. In particular, the adversarial process uses two models: i) a generative model $G$, and ii) a discriminative model $D$ that are trained playing a zero-sum game. Following this general framework, [7] proposed a single image super-resolution architecture called SRGAN. Although this approach is unsupervised, part of its loss is still supervised. In fact, a content loss term based on a per-pixel loss between the output and ground-truth images is used there. This term requires again alignment between LR and HR and limiting its applicability in our context. Another drawback of SRGAN is its difficulty to train, often generating SR images that are too sharp or have artifacts. To reduce these drawbacks, [4] proposed to combine a VB approach with GAN. They show that an asymmetric loss function obtained using a cross-entropy loss for the discriminative network and a mean discrepancy objective for the generative network, make the GAN training more stable. Similarly to this idea, an Adversarial Variational Bayes was proposed by [9] where a Variational Autoencoder (VAE) is trained using an auxiliary discriminative network. Contrary to the previous case, this approach provides a more clear theoretical justification. However, the problem of using paired LR/HR has not been resolved by any of the approaches described so far. One of the first approaches that formalize the possibility to translate images from a source domain $X$ to a target domain $Y$ in the absence of paired examples was proposed by [19] and is called CycleGAN. Using an adversarial training the goal of this method is to learn a mapping $G : X \rightarrow Y$ such that the distribution of images from $G(X)$ is indistinguishable from the distribution $Y$. Since this mapping is highly under-constrained, the authors also introduced an inverse mapping $F : Y \rightarrow X$ and a cycle consistency loss to push $F(G(X)) \approx X$. Thanks to this two-step consistency, the need for the paired images is eliminated. Varying the input-output domain, this framework can be used to perform artistic style transformation [6] (where, for example, a horse can be converted into a zebra) or, as in our case, transfer the resolution from one domain to another.

Another interesting approach was proposed by [14]. Here the authors question that the predetermined LR images obtained from standard bi-cubic down-sampling rarely look like the real LR images. So they introduce a method called Zero-Shot SR, that does not rely on prior training. To do so they exploit the internal recurrence of information inside a single image and train a small image-specific CNN at test time. This facilitates self-training SR for biological data, old photos, noisy images, and other images where the acquisition process is unknown.

Following the idea proposed by [19] and [14] we propose an unsupervised framework that uses unpaired images and that can be used in the case where aligned pairs of LR/HR images are missing.

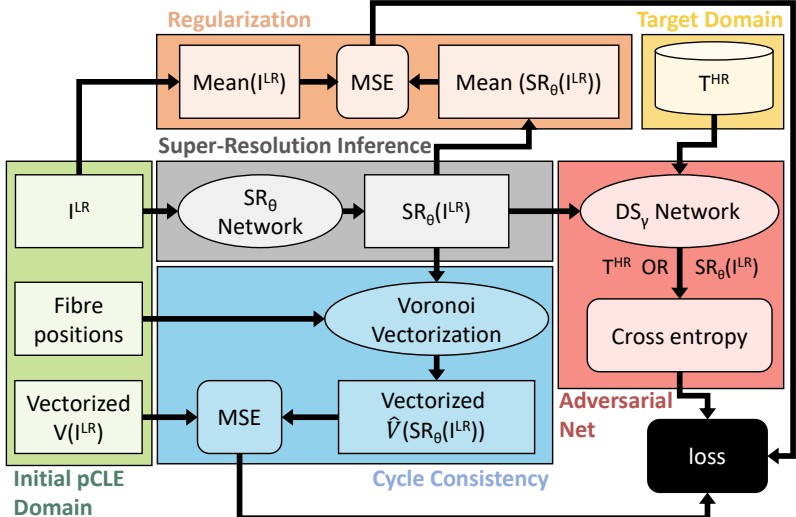

Figure 2: Pipeline used for training the proposed adversarial network with cycle consistency. Each component of the pipeline is identified by a different color.

## 3 Materials and methods

### 3.1 Database

To validate our solution, we used the database proposed by [1] containing 238 anonymized probe-based Confocal Laser Endomicroscopy (pCLE) video sequences of the colon and oesophagus. This database does not provide the real ground truth of the HR images. Only estimated $\widehat{HR}$, computed using a video-registration technique on the LR images are available here.

The database was divided into three subsets: a train set (70%), a validation set (15%), and a test set (15%). Images from each clinical setting were distributed equally in the three subsets. After following the same data-normalization proposed by [12], we used non-overlapping patches of 64×64 pixels extracted from the train and validation set for training purposes. The patches in the validation set were used to monitor the loss and avoid overfitting. Full-size images from the test set were instead processed to compute the final results. The full-size processing of the test images is possible since the inference network is fully convolutional and no specific image size is required as input.

### 3.2 Adversarial training

The pipeline used for training our framework is presented in Fig. 2 and it is divided into different sub-sections, each coded by a specific colour.

We formalize our training as an adversarial min-max problem where two networks, a discriminative network defined as $DS_\gamma$ (red sub-sequence in Fig. 2), and a super-resolution network defined as $SR_\theta$ (grey sub-sequence in Fig. 2) are trained simultaneously. More specifically, the first network $DS_\gamma$ is trained solving:

$$\max_{\gamma} \mathbb{E}_{x \sim p_{I_{LR}}} \big[ log \big( 1 - DS_\gamma \big( SR_\theta(x) \big) \big) \big] + \mathbb{E}_{y \sim p_{T_{HR}}} \big[ log DS_\gamma(y) \big], \tag{1}$$

where $p_{I_{LR}}$ and $p_{T_{HR}}$ are respectively the patch distributions on the input and target domain, $DS_\gamma(*)$ gives the probability that a patch comes from the target domain, whereas $SR_\theta(x)$ is the predicted super-resolved patch obtained from $x$. The meaning of Eq.1 is that the discriminator has to maximize how to discriminate predicted super-resolved images from real $T^{HR}$ patches.

The second network $SR_\theta$, is trained instead through the minimization of a composite loss function $loss_t$ obtained solving:

$$\min_\theta \mathbb{E}_{x \sim p_{I^{LR}}} \big[loss_t\big(x, SR_\theta(x)\big)\big] \tag{2}$$

The proposed $loss_t$, defined in Eq. 3, is a combination of three terms: $l_{Vec}$ that models the physical acquisition characteristics of the predicted super-resolved patch, $l_{Adv}$ that models the adversarial loss function and $l_{Reg}$ used to regularize the network training. The details of each term are provided later in this section.

$$loss_t = l_{Vec} + l_{Adv} + l_{Reg} \tag{3}$$

Both $SR_\theta$ and $DS_\gamma$ are simultaneously trained using the back-propagation algorithm that gradually adjusts the parameters $\theta$ and $\gamma$ through a stochastic gradient descent for the former and a stochastic gradient ascent for the latter.

### 3.3 Input domain and cycle consistency

#### 3.3.1 Input domain

The green blocks in Fig. 2 represent the data structures required as input for the proposed pipeline. The first input is the reconstructed $I^{LR}$ that is used by the $SR_\theta$ network to infer the super-resolved patch $SR_\theta(I^{LR})$.

In the pCLE imaging, image acquisition is achieved by illuminating one fibre at a time. Each fibre acts as an individual pinhole and a scan point for fibre confocality. The information from all the points is then collected in a vector that we defined as vectorized image $V(I^{LR})$, and represents another input block in our pipeline. $I^{LR}$ images are reconstructed interpolating the values in $V(I^{LR})$ from the centres of the fibre positions to the points of a regular grid. Therefore the fibre positions are the last input block required by our pipeline.

#### 3.3.2 Cycle consistency

Starting from a generated high-resolution pCLE image $SR_\theta(I^{LR})$, we can obtain a low-resolution representation of it, by a process referred to as Voronoi vectorization $\widehat{V}(SR_\theta(I^{LR}))$ which is equivalent to the down-sampling for standard images.

The vectorized $V(I^{LR})$ and the Voronoi vectorization $\widehat{V}(SR_\theta(I^{LR}))$ are used in our pipeline to create the cycle consistency (blocks coloured in cyan in Fig. 2). These blocks are used to impose the requirements for the predicted super-resolved images $SR_\theta(I^{LR})$ to have the same physical acquisition properties as the initial $I^{LR}$ images. Without this cycle consistency, the network could simply produce arbitrary images in the target domain with no relationship to the structures contained in the input image, because our framework relies on unpaired patches. To avoid this, we force the $V(I^{LR})$ and $\widehat{V}(SR_\theta(I^{LR}))$ to be similar using the $l_{Vec}$ term in the proposed loss function.

$$l_{Vec} = \frac{1}{682} \sum_{i=1}^{682} \left[ V(I^{LR})_i - \widehat{V}(SR_\theta(I^{LR}))_i \right]^2 \tag{4}$$

The details of the Voronoi vectorization used in our framework are described in Fig. 3. Here, the first step is to compute the Voronoi diagram from the fibre positions. The result is a partition of the plane where for each fibre there is a corresponding region, called Voronoi cell, consisting of all points closer to it than to any other fibre. The next step is to average the pixels in the $SR_\theta(I^{LR})$ patch that belong to the same Voronoi cell, imitating the point spread function of the fibre acquisition process. Since each patch may have a different number of fibres, the vectorization can produce vectors with different sizes. Therefore, a 0-padding is introduced so that each vector always have 682 elements. As a final step, all the elements in the vector are normalized in the range [0, 1]. This normalization makes the training faster and reduces the chances of getting stuck in local optima.

In contrast to CycleGan [19], our cycle consistency block is not a trainable network, but rather is used to constrain the $SR_\theta$ network to generate images with the same physical acquisition properties as the initial $I^{LR}$ images.

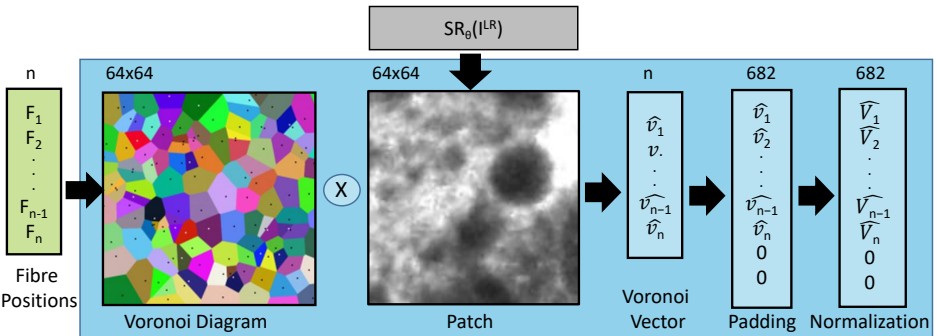

Figure 3: Voronoi vectorization used in our pipeline to constrain the predicted super-resolved patches.

## 3.4 Super-resolution network

We decided to use the layout for the SR network propose in [7]. $SR_\theta$ is aimed at producing images that are similar to the one in the target domain by trying to fool the discriminator network. This is achieved through the term $l_{Adv}$ in the proposed loss function defined as follows:

$$l_{Adv} = -log DS_\gamma(SR_\theta(I^{LR})) \tag{5}$$

where $DS_\gamma(SR_\theta(I^{LR}))$ is the probability that the predicted image $SR_\theta(I^{LR})$ is classified as a real $T^{HR}$. In the inference phase, only $SR_\theta$ is used for processing the $I^{LR}$ images.

## 3.5 Regularization

The blocks displayed in orange in Fig. 2 is used to regularize the network training. This regularization is required since the Voronoi vectorization of each patch is normalized to the range [0-1] and this may result in an expansion of its histogram range. To restore the correct histogram distribution, we impose that the mean values in each row and each column of the patch are identical between the initial $I^{LR}$ and the obtained $T^{HR}$. This is achieved in our framework through the $l_{Reg}$ term of $loss_t$:

$$l_{Reg} = \frac{1}{H}\sum_{y=1}^{H}\left[\frac{1}{W}\sum_{x=1}^{W}SR_\theta(I_{xy}^{LR}) - \frac{1}{W}\sum_{x=1}^{W}I_{xy}^{LR}\right]^2 + \frac{1}{W}\sum_{x=1}^{W}\left[\frac{1}{H}\sum_{y=1}^{H}SR_\theta(I_{xy}^{LR}) - \frac{1}{H}\sum_{y=1}^{H}I_{xy}^{LR}\right]^2 \tag{6}$$

## 3.6 Target domain

In our pipeline we considered four different target domains to transfer the super-resolution to the LR images: i) $T_{nat}^{HR}$ where the HR patches are extracted from natural images (gray-scaled images from the Sun2012 database [18]), ii) $T_{unpair}^{HR}$ containing unpaired HR patches obtained by a video-registration technique on the LR images, iii) $T_{pair}^{HR}$ containing paired HR patches obtained using a video-registration technique while the LR are synthetically aligned, and iv) $T_{res}^{HR}$ where the HR patches are obtained by down-sampling the LR images by a factor of four. The idea behind this last target domain is that patches in the images have recurrences at a different scale and down-sampling large LR images may increase the high-frequency responses in the generated down-sampled HR patches.

## 3.7 Training details and parameters

In our implementation, Eq 1 is solved by minimizing the cross-entropy of the number of samples correctly discriminated by $DS$. As proposed by [2] we add white noise to the inputs of the $DS_\gamma$ network to stabilize the adversarial training. We trained our networks on an NVIDIA GTX TITAN-X GPU card with 12 GB of memory. The training procedure converges after 50-80 thousand iterations

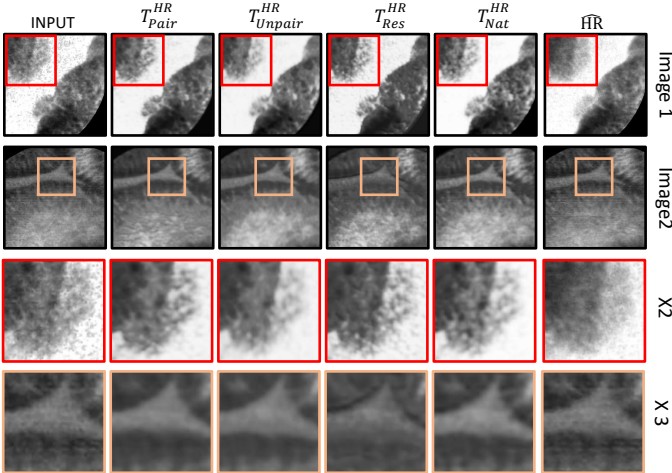

Figure 4: Example of visual results obtained by the proposed approaches when trained with different target domains. From left to right we have: Input, training with $T^{HR}_{pair}$, training with $T^{HR}_{unpair}$, training with $T^{HR}_{res}$, training with $T^{HR}_{nat}$ and $\widehat{HR}$.

of random mini-batch with 54 patches. For the optimization of the stochastic gradient descent, we use Adam with $\beta1 = 0.9$, $\beta2 = 0.999$ and $\epsilon = 10e\text{-}8$. The networks were trained with a learning rate of 10e-4.

## 4 Experiments

Due to the lack of real ground truth in our database, the validation of our experiments is based on complementary quantitative and qualitative analysis. The quantitative analysis, presented in Section 4.1, uses four different metrics to evaluate the obtained images. The qualitative analysis is instead based on a MOS study carried out by clinicians and medical imaging experts that gave numerical indications of the perceived quality of the super-resolved images.

### 4.1 Quantitative analysis

The four metrics used in our quantitative analysis are: i) a Structural Similarity matrix (SSIM) proposed by [17] that evaluates the similarity between $SR_\theta(I^{LR})$ and $\widehat{HR}$, ii) $\Delta GCF_{\widehat{HR}}$ that quantifies the improvement on the global contrast factor (a reference-free metric for measuring image contrast [8]) that the super-resolved image yields with respect to $\widehat{HR}$, iii) $\Delta GCF_{I^{LR}}$ that is the improvement of the global contrast factor that the super-resolved image yields with respect to the initial $I^{LR}$, and iv) a composite score $Tot_{cs}$ obtained by averaging the normalized value of SSIM with the normalized value of $\Delta GCF_{\widehat{HR}}$. This composite score leads to a more robust evaluation of the results since, SSIM alone is not reliable when the ground truth is only estimated, while the GCF can be improved by merely adding random high frequency to the images.

Our first experiment is aimed at finding the best target domain for improving the pCLE images. These results are reported in Table 1 and, as we can see, the network trained with natural images obtain the best $Tot_{cs}$ score. We can also see that training our network with paired images performs worse than training it with the unpaired one. This is probably due to the fact that the paired images are obtained synthetically and they may have a wide domain gap with the real images. With this result, we can state that paired patches are not anymore a requirement for our framework. Finally, re-scaling LR images to a low scale does not seem to provide good results and the high-frequency signals are not recovered. These qualitative indications can be seen on reconstructed images reported in Fig. 4.

To further validate our framework, we compare our best approach (the network trained with the target domain $T^{HR}_{nat}$), against some state-of-the-art single image super-resolution methodologies. These results are presented in Table 2. In this experiment we consider three different approaches: i) the

Table 1: Quantitative analysis results obtained by our approach when trained with different target domains

| | $T_{pair}^{HR}$ | $T_{unpair}^{HR}$ | $T_{res}^{HR}$ | $T_{nat}^{HR}$ |
|---|---|---|---|---|
| $SSIM_{\widehat{HR}}$ | **0.91 ± 0.03** | **0.91 ± 0.03** | 0.88 ± 0.04 | 0.87 ± 0.03 |
| $\Delta GCF_{\widehat{HR}}$ | 0.01 ± 0.29 | 0.38 ± 0.27 | -0.13 ± 0.39 | **0.66 ± 0.31** |
| $\Delta GCF_{I_{LR}}$ | -0.28 ± 0.18 | 0.09 ± 0.19 | -0.41 ± 0.31 | **0.37 ± 0.26** |
| $Tot_{cs}$ | 0.53 | 0.64 | 0.45 | **0.66** |

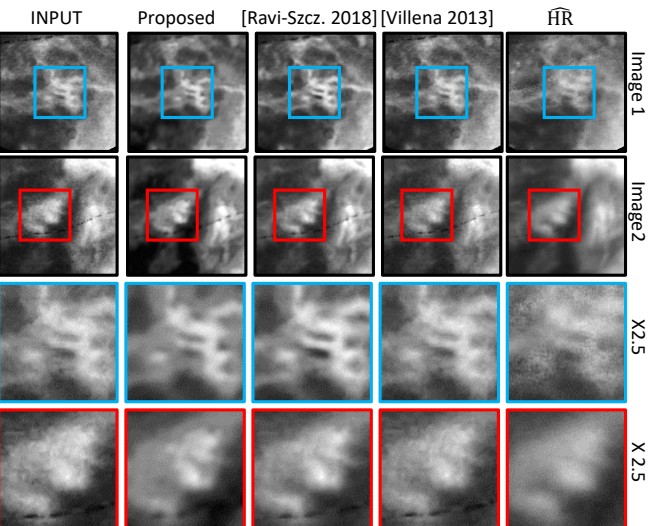

Figure 5: Example of visual results obtained by our approach in comparison with other state-of-the-art approaches. From left to right we have: Input image, proposed output, output from [12], output from [16] and $\widehat{HR}$.

unsupervised Wiener deconvolution tuned on the train set, ii) the unsupervised variational Bayesian inference approach with sparse and non-sparse priors [16], and the supervised EBSR proposed by [12]. Finally, a contrast-enhancement approach obtained by sharpening the input was also used as a baseline. Although the sharpening algorithm produces the best contrast improvements, our approach is the one that obtains the highest SSIM and, according to $Tot_{cs}$, the overall performance outperforms all the other approaches. The statistical significance of these improvements was assessed with a paired t-test (p-value less than 0.0001). Visual examination for some of these images shown in Fig. 5, confirms these quantitative results.

## 4.2 Semi-quantitative analysis (MOS)

To perform the MOS, we asked 10 trained individuals to evaluate, on average, 20 images each. At each step, the SR images obtained with the proposed approach, with [12], and with a contrast-enhancement approach (baseline) that sharpens the input, were shown to the user in a random order. The input and the $\widehat{HR}$ were also displayed on the screen as references for the participants. For each of the three images, the user assigned a score between 1 (strongly disagree) to 5 (strongly agree) on the following questions:

- *Q1: Is the image artefact-free?*
- *Q2: Can you see an improvement in contrast with respect to the input?*
- *Q3: Can you see an improvement in the details with respect to the input?*
- *Q4: Would you prefer seeing the new image over the input?*

To make sure that the questions were correctly interpreted, each participant received a short training before starting the study. The results on the MOS presented in Fig. 6 show that the proposed approach

Table 2: Quantitative analysis results of the proposed approach against state-of-the-art methods

| | Proposed | Ravi-Szcz. [2018] | [Villena 2013] | Wiener | Contrast-enhancement |
|---|---|---|---|---|---|
| $SSIM_{\widehat{HR}}$ | $\mathbf{0.87 \pm 0.03}$ | $0.86 \pm 0.04$ | $0.87 \pm 0.05$ | $0.84 \pm 0.06$ | $0.61 \pm 0.08$ |
| $\Delta GCF_{\widehat{HR}}$ | $0.66 \pm 0.31$ | $0.41 \pm 0.23$ | $0.27 \pm 0.21$ | $-0.01 \pm 0.31$ | $\mathbf{1.34 \pm 0.36}$ |
| $\Delta GCF_{ILR}$ | $0.37 \pm 0.26$ | $0.13 \pm 0.12$ | $-0.01 \pm 0.00$ | $-0.30 \pm 0.18$ | $\mathbf{1.05 \pm 0.25}$ |
| $Tot_{cs}$ | $\mathbf{0.66}$ | $0.59$ | $0.55$ | $0.43$ | $0.53$ |

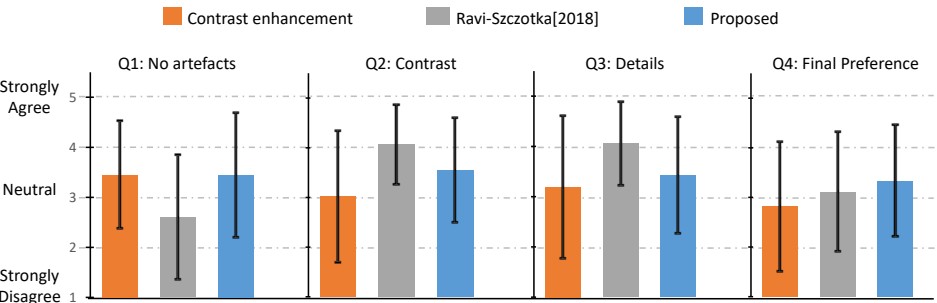

Figure 6: Mean and standard deviation of the participants' replies to each of the four MOS questions for the evaluation of the following approaches: contrast-enhancement (baseline), [12], and proposed.

and [12] provide complementary features. In fact, although the details (question Q3) and the contrast (question Q2) in our approach seem to be worse than [12], our solution provides a better score for the absence of artefacts (question Q1), which is an important characteristic in clinical applications. Regarding the final preferred approach (question Q4), our solution shows the best results [12], confirming the validity of our approach to perform super-resolution on pCLE images. The approach that sharpens the images is, instead, the one that provides the lowest scores for Q2, Q3 and Q4, probably because it enhances the noise.

## 5 Discussion and conclusions

We report a super-resolution framework for endomicroscopy images based on an unsupervised adversarial deep neural network that takes advantage of the knowledge of the physical acquisition process to impose a cycle consistency. The proposed framework results to be particularly useful in all situations where there is a lack of HR images and pairs of LR/HR images are not available for the supervised training.

To the best of our knowledge, we are the first to propose an unsupervised super-resolution approach for medical images. Our results validated using a database of 238 endomicroscopy video sequences show the ability of the pipeline to produce convincing super-resolved images. Further clinical trials could validate the relevance of the proposed framework to specific clinical applications for super-resolution.

## Acknowledgement

**Funding:** This work was supported by Wellcome/EPSRC [203145Z/16/Z; NS/A000050/1; WT101957; NS/A000027/1; EP/N027078/1]. This work was undertaken at UCL and UCLH, which receive a proportion of funding from the DoH NIHR UCLH BRC funding scheme. The PhD studentship of Agnieszka Barbara Szczotka is funded by Mauna Kea Technologies, Paris, France.

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
