# OpenReview forum: "Adversarial training with cycle consistency for unsupervised super-resolution in endomicroscopy"
_MIDL.amsterdam/2018/Conference — MIDL 2018 Oral_

### Review · AnonReviewer2 · 2018-05-04
**Very well done!**

**Rating:** 5
**Confidence:** 2

**Review:**

The paper describes a method to create super-resolution images from low-resolution images using a generative adversarial network approach that does not require paired low-res/hi-res images.

This was a very pleasant read and very informative. The related work is relevant and adds to the paper. The methods of this paper are well described and strongly motivated. The results are presented in a thorough and objective manner, both quantitatively and qualitatively. I would like to see more of a discussion section to put this work in a larger context at the end, but the absence might be space limited.

I would encourage the authors to add some high level talking points surrounding the practical use of the method in the discussion section, depending on available space; I would not recommend cutting down on any other portion of the paper, so this could be the limiting factor.
Some interesting talking points to this method could include:
1) Without a high-resolution analog for validation, in the real world is there a risk of the algorithm "making up" data or structures?
2) More and more algorithms are interpreting images as opposed to humans, does super-resolution have benefits to algorithmic interpretation or is it limited to human interpretation?
3) Are there any examples of super-resolution images changing a reader's interpretation or the speed of interpretation? Or are do they simply lead to a more confident read?

The fact that the paper leads to these thought provoking questions is a testament to the strength of the paper. Well done.

**Special Issue:**

Definitely

---

### Review · AnonReviewer3 · 2018-05-09
**Review of Adversarial training with cycle consistency for unsupervised super-resolution in endomicroscopy**

**Rating:** 3
**Confidence:** 2

**Review:**

The authors investigated unsupervised SR in endomicroscopy. This can exploit HR images, regardless of the domain where they are coming from, to transfer the super-resolution to the initial LR images. This property can be particularly
useful in all situations where pairs of LR/HR are not available during the training. This study looks very challenge and very important and experimental results are relatively extensive.

**Special Issue:**

Yes

---

### Review · AnonReviewer1 · 2018-05-09
**A well written and detailed paper on a relevant topic**

**Rating:** 5
**Confidence:** 2

**Review:**

The paper describes a method to improve image resolution in endomicroscopy using only unpaired training samples and an adversarial framework.  This is a highly relevant topic since paired samples are difficult to obtain in many applications in medical imaging and the adversarial network has been exploited to excellent effect for this purpose in the natural image domain.  Aspects of image acquisition are used to constrain the network and enforce cycle consistency.
Overall I found this paper to be extremely relevant, well-written and clearly explained.  The technique is novel and the current state of the art is well described.  Experiments are described in good detail and are thorough in their construction.  While the authors do not have a gold-standard available they make every effort to provide quantitative and qualitative evaluation of their results and demonstrate an improvement compared to the state of the art.  It would be a nice addition to test the method in a domain where both HR and LR images are available so that a true gold standard could be utilised - however I realise that this may not be easy to achieve depending on access to data.
In conclusion I am very happy to recommend this paper for acceptance.

**Special Issue:**

Yes

---

### Comment · ~Bram_van_Ginneken1 · 2018-05-18
**Selection for longlist for special issue Medical Image Analysis**

Dear authors,

Congratulations on your acceptance to MIDL! We have selected your paper on the longlist for the Medical Image Analysis Special Issue. Please read this page:
https://midl.amsterdam/special-issue-in-medical-image-analysis/
Please answer the three questions that are listed on that page about your interest in submitting to the special issue, potential overlap with other publications, and related publications.

You can post your answer here directly below on openreview.net, or mail me directly at bram.vanginneken@radboudumc.nl.

Best regards, Bram

---

> ### Comment · ~Daniele_Ravi1 · 2018-05-21
> **Answers regarding our interest in the MedIA special issue**
>
> 1)We confirm our strong interest in submitting our work to the MedIA special issue. To this end, we are eager to augment the content on the manuscript significantly. As suggested by the reviewers, we will extend the manuscript showing that, although there are no real matched pairs of input images and ground-truth high-resolution images for validation, the risk of obtaining spurious changes in the images structures or compromising the existing details is limited. On the other hand, we will emphasize the benefits in using super-resolution algorithms and why they can lead to a better medical interpretation (for example, by reviewing clinical contexts where they have been successfully used for this purpose).
>
> 2)Yes, we confirm that the paper or any other paper with overlap work is not under review or consideration elsewhere.
>
> 3)A related initial approach was developed in the paper D. Ravì*, A. B. Szczotka*, D. I. Shakir, S. P. Pereira, and T. Vercauteren. Effective deep learning training for single-image super-resolution in endomicroscopy exploiting video-registration based reconstruction. IJCARS (in press), March 2018.
> The main limitation of this preliminary work is that the method requires aligned pairs of low-res/high-res images. Since these pairs do not exist for our clinical application, synthetic pairs were generated by exploiting a video-registration technique (used to estimate the HR images from a sequence of LR images) and a pipeline that models the acquisition process to generate back synthetic LR data.
> In the work accepted for presentation at MIDL, our new pipeline relaxes the constraints required by standard super-resolution algorithms and no matched pairs are required. We propose instead an unsupervised SR framework based on an adversarial network, equipped with a physically-inspired cycle consistency designed to impose directly some acquisition properties on the super-resolved images. Our framework can exploit HR images, regardless of the domain where they are coming from, to transfer the super-resolution to the initial LR images.

---

### Decision · Program_Chairs · 2018-05-15
**Paper48 Acceptance Decision**

Oral